# Integrating Metabolomics and Network Analyses to Explore Mechanisms of *Geum japonicum* var. *chinense* Against Pulmonary Fibrosis: Involvement of Arachidonic Acid Metabolic Pathway

**DOI:** 10.3390/ijms26041462

**Published:** 2025-02-10

**Authors:** Junyan Ran, Qian Wang, Tao Lu, Xiuqing Pang, Shanggao Liao, Xun He

**Affiliations:** 1State Key Laboratory of Functions and Applications of Medicinal Plants, School of Pharmaceutical Sciences, Guizhou Medical University, Gui’an New District, Guiyang 561113, China; rjy123000@163.com (J.R.); wq112240920@163.com (Q.W.); lutao18212430505@163.com (T.L.); pangxiuqing@163.com (X.P.); 2University Engineering Research Center for the Prevention and Treatment of Chronic Diseases by Authentic Medicinal Materials in Guizhou Province, Gui’an New District, Guiyang 550025, China; 3Engineering Research Center for the Development and Application of Ethnic Medicine and TCM, Ministry of Education, Guiyang 550004, China; 4Guizhou Provincial Engineering Technology Research Center for Chemical Drug R&D, Guizhou Medical University, Guiyang 550014, China

**Keywords:** *Geum japonicum* var. *chinense*, pulmonary fibrosis, metabolomics, network analysis, AA metabolism

## Abstract

Pulmonary fibrosis (PF) emerges as a significant pulmonary sequelae in the convalescent phase of coronavirus disease 2019 (COVID-19), with current strategies neither specifically preventive nor therapeutic. *Geum japonicum* var. *chinense* (GJC) is used as a traditional Chinese medicine to effectively treat various respiratory conditions. However, the protective effects of GJC against PF remains unclear. In the present study, the anti-PF effect of GJC aqueous extract was studied using a PF mouse model induced by bleomycin (BLM). To characterize the metabolite changes related to PF and reveal therapeutic targets for GJC aqueous extract, we performed metabolomic and network analysis on mice lungs. Finally, key targets were then validated by Western blotting. GJC aqueous extract effectively alleviated the onset and progression of lung fibrosis in PF mice by inhibiting inflammatory responses and regulating oxidative stress levels. Integrating serum metabolomics and network analyses showed the arachidonic acid (AA) pathway to be the most important metabolic pathway of GJC aqueous extract against PF. Further validation of AA pathway protein levels showed a significant rise in the levels of ALOX5, PTGS2, CYP2C9, and PLA2G2A in PF lungs. GJC aqueous extract treatment regulated the above changes in metabolic programming. In conclusion, GJC is a promising botanical drug to delay the onset and progression of PF mice. The primary mechanism of action is associated with the comprehensive regulation of metabolites and protein expression related to the AA metabolic pathway.

## 1. Introduction

Pulmonary fibrosis (PF) is a chronic, progressive, and fibrotic interstitial lung disorder characterized by dysregulated proliferation of fibroblasts, abnormal accumulation of the extracellular matrix (ECM), inflammatory injury, and subsequent functional decline in lung tissue [1]. Clinically, PF progresses with morphological alterations in lung architecture and compromised lung function, and ultimately leads to respiratory failure or death. The median survival for IPF patients is approximately 3–5 years. Increasing evidence suggests that severe COVID-19 patients may suffer persistent lung alterations leading to the development of PF, posing a considerable public health threat [2,3]. Although considerable research has been dedicated to PF, the complexity of its pathogenesis remains enigmatic, with a multifactorial etiology involving immune responses, inflammation, epithelial–mesenchymal transition (EMT), and oxidative stress [4]. The therapeutic landscape for PF is currently limited, with lung transplantation as the sole viable treatment for advanced cases with respiratory failure. The International Guidelines for Pulmonary Fibrosis (2015) endorse the use of pirfenidone (PFD) and nintedanib; however, these medications are costly and are associated with unfavorable prognoses [5,6]. Thus, there is an urgent demand for the development of efficacious treatments for PF.

In recent years, traditional Chinese medicine (TCM) has made significant strides in preventing and treating complex diseases such as liver, kidney, and lung fibrosis [7,8]. *Geum japonicum* var. *chinense* (GJC) is a well-regarded TCM for its efficacy in invigorating qi and spleen, nourishing blood and yin, and moistening lungs for phlegm resolution [9]. Modern pharmacological studies have highlighted GJC’s anti-inflammatory and anti-tumor activities [10,11]. Clinically, GJC serves as a key ingredient in “Fenghanmaogan Capsules” and “Shengning Cold Liquid”, used in the treatment of inflammatory respiratory ailments. Given that PF is also a lung disease closely associated with inflammation, GJC may hold promise in the therapy of PF.

As a traditional Chinese medicines (TCM), GJC is chemically complex, with undefined bioactive components and unidentified therapeutic targets. Network pharmacology, which integrates multi-compound-based pharmacology with bioinformatics, has achieved considerable success in analyzing the potential effective compounds and the molecular mechanisms of TCMs [12]. Upon administration, TCMs release pharmacologically active substances as prototype components or metabolites into the serum, leading to expression-level alterations of the organism’s endogenous metabolites. Metabolomics, focusing on comprehensive analysis of these metabolic alterations, has demonstrated its potential in assessing the efficacy of TCMs, elucidating their pharmacologically active constituents, and exploring their mechanisms of action [13,14].

In this study, the anti-PF effect of GJC aqueous extract was studied using a PF mouse model induced by bleomycin (BLM) and its mechanism was investigated by using a serum metabolomics method. Network pharmacology was then analyzed on the components of GJC identified by Ultra Performance Liquid Chromatography-Quadrupole-Exactive Mass Spectrometry (UPLC-Q-Exactive MS) to show the potential targets and pathways involved in the anti-PF effect of GJC aqueous extract. Integrated analysis of serum metabolomics and network pharmacology was conducted to reveal the bioactive metabolites and mechanism. Key targets were then validated and analyzed.

## 2. Results

### 2.1. Preliminary Identification of Compounds in GJC Aqueous Extract

The base peak chromatograms (BPCs) of GJC in both positive and negative ion modes of Ultra-High Performance Liquid Chromatography-Quadrupole-Orbitrap High Resolution Mass Spectrometry (UHPLC-Q-Orbitrap HRMS) are presented in Figure 1. By searching the Compound Discoverer 2.1 database and consulting the relevant literature, a total of 26 compounds (Table 1) were identified based on accurate molecular mass and secondary fragment ions. Isomers were assigned on the basis of their relative retention times, indicated by their calculated logP. The structures of these compounds are provided in Appendix A.

### 2.2. The Effects of GJC Aqueous Extract on the Survival Rate and Lung Coefficient in BLM-Induced PF Mice

As shown in Figure 2A, BLM significantly decreased the survival rate of mice (*p* < 0.01 vs. Control), whereas PFD and GJC aqueous extract treatments markedly improved the survival rates. Meanwhile, lung coefficients (*p* < 0.01) were significantly increased after BLM induction, but were remarkably decreased by PFD and GJC aqueous extract treatments (*p* < 0.05 or <0.01)

### 2.3. H&E Staining and Immunohistochemical Staining Analysis

Hematoxylin and Eosin (H&E) staining analysis indicated that the pathological score (Figure 3C) of lung tissue sections from mice in the BLM group was higher than that in the Control group, as evidenced by thickened alveolar walls, partial disappearance of alveolar spaces, extensive proliferation of lung interstitium, abundant infiltration of inflammatory cells within the interstitium, and observable degeneration of bronchial epithelial cells (Figure 3A). In comparison to the BLM group, GJC-treated groups exhibited varying degrees of alleviation of lung tissue structural damage. Additionally, Masson staining (Figure 3B,F) revealed a significant increase in blue-stained collagen fibers in the BLM group (vs. the Control group). However, following the administration of GJC aqueous extract, there was a substantial reduction in collagen fiber deposition, suggesting potential ameliorative effects of GJC aqueous extract on PF-related pathological damage in mice. In the fibrosis process of PF, Transforming Growth Factor-β1 (TGF-β1) can stimulate the transformation of fibroblasts into myofibroblasts and promote the expression of α-smooth muscle actin (α-SMA). Meanwhile, the high expression of α-SMA further confirms the activation and proliferation of myofibroblasts, thereby exacerbating the process of fibrosis. Therefore, immunohistochemical (IHC) analysis is employed to detect the expression levels of TGF-β1 and α-SMA in lung tissue (Figure 3C,D,G,H). The results demonstrate significantly higher levels of TGF-β1 and α-SMA proteins in the lung tissue of the BLM group compared to the Control group (*p* < 0.05 or *p* < 0.01). In contrast, the lung tissue of the PFD group and the groups treated with GJC shows varying degrees of reduction in the levels of the two proteins (*p* < 0.05 or *p* < 0.01) (Figure 3G,H).

### 2.4. The Effect of GJC Aqueous Extract on the HYP Level in Lung Tissues of PF Mice

Hydroxyproline (HYP), as the primary constituent of collagen, plays a crucial role in the synthesis and stabilization of collagen, and its level in the lung tissue indirectly reflects the degree of fibrosis in the lung and therefore serves as a diagnostic biomarker for PF. The results (Figure 4A) demonstrate that BLM significantly increased the HYP level (*p* < 0.001) in the Control group, while treatments with PFD and GJC aqueous extract significantly decreased the level (*p* < 0.01 or *p* < 0.001 vs. BLM group). These findings suggested that GJC could significantly suppress the development of PF.

### 2.5. Effect of GJC Aqueous Extract on Inflammatory Levels in Serum of BLM-Induced PF Mice

Inflammation plays a crucial role in the pathogenesis of IPF, serving as both a key initiating factor of the disease and an important driver of fibrosis progression [25]. To evaluate the impact of the GJC aqueous extract on BLM-induced inflammation in mice, ELISA was utilized to detect the levels of IL-6 and TNF-α in mouse serum. The results (Figure 4B,C) indicate that compared to the serum levels of IL-6 and TNF-α in the Control group, those in the BLM group were significantly elevated (*p* < 0.05 or *p* < 0.01). Meanwhile, treatments with PFD and different doses of GJC aqueous extract all significantly reduced inflammatory levels.

### 2.6. The Effect of GJC Aqueous Extract on Oxidative Stress in the Lung Tissue of PF Mice

Oxidative stress responses are known to exacerbate the progression of lung fibrosis. Superoxide dismutase (SOD) is an antioxidant enzyme playing a crucial role in mitigating oxidative stress, while malondialdehyde (MDA) is widely accepted to be a critical biomarker of oxidative stress. To evaluate the impact of GJC aqueous extract on oxidative stress in the context of PF, the levels of SOD and MDA in the lung tissue of the mice were determined. The results (Figure 4D,E) demonstrate that BLM significantly reduced the SOD levels (*p* < 0.01 vs. Control group) and increased the MDA levels (*p* < 0.01 vs. Control group) of mice; conversely, treatments with PFD and GJC significantly increased the SOD levels and decreased the MDA levels (*p* < 0.05 or *p* < 0.01 vs. BLM group). These findings suggest that the reduction in oxidative stress was involved in the anti-PF effect of GJC aqueous extract.

### 2.7. Metabolomics Analysis Identified Differential Metabolites and Key Pathways

To identify potential PF biomarkers regulated by GJC aqueous extract, OPLS-DA analysis was performed on the serum UHPLC-Q-Exactive Orbitrap MS data using SIMCA14.1 software (Figure 5A,B). The results show that the established OPLS-DA model was not over-fitted (Appendix A). Meanwhile, a total of 32 differential metabolites with VIP > 1 and *p* < 0.05 in the S-plots (Appendix A) were identified between samples of different groups. Among these, 21 metabolites were identified in positive ion mode, and 11 in negative ion mode. Cross-validation of these metabolites was conducted using the HMDB database, and relevant information for these metabolites is provided in Appendix A. To visualize the expression patterns of differential metabolites among different sample groups, a heatmap analysis was conducted. Additionally, bar charts were used to visualize the expression of selected differential metabolites with significant regulation (Figure 6A,B). The results indicate that 32 differential metabolites either downregulated (5) or upregulated (27) in the BLM group were reversely regulated by GJC aqueous extract, suggesting that GJC aqueous extract has the potential to effectively alleviate the metabolic disturbances caused by BLM-induced PF in mice. Metabolic pathway enrichment analysis on the 32 differential metabolites showed a total of 10 metabolic pathways (Figure 6C) including AA metabolism, vitamin B6 metabolism, and sphingolipid metabolic pathways. Notably, AA metabolism, with *p* < 0.05 and impact > 0.1, was the most important metabolic pathway involved in the effect of GJC aqueous extract on PF.

### 2.8. Network Analysis Predicted the Potential Targets and Pathways of GJC Aqueous Extract Against PF

The therapeutic effects of GJC aqueous extract on PF and its mechanism of action were also investigated by network analysis. Initially, Swiss Target Prediction and PharmMapper databases were queried to compile a list of 734 targets associated with the 26 identified compounds of GJC aqueous extract. Subsequently, 1270 PF-related disease targets were sourced from Genecards, OMIM, and Drugbank databases (Figure 7A). A total of 153 core targets associated with both GJC and PF were imported into STRING for PPI to obtain the target protein association network (screened by confidence score > 0.9). The PPI network constructed with Cytoscape 3.9.1 had 122 nodes and 368 edges (Figure 7B). The strength of interactions between targets was represented by the thickness of connecting lines in the PPI graph. Degree values of these targets, calculated using the CytoNCA 2.1.6 plugin, reflected their centrality in the network. Further analysis involved Gene Ontology (GO) and Kyoto Encyclopedia of Genes and Genomes (KEGG) enrichment of the 122 PPI targets (Figure 7C–E). Gene Ontology (GO) analysis of the 122 PPI targets revealed enrichment of positive regulation of protein kinase B signaling, MAPK cascade, inflammatory response in the biological processes, extracellular region, cytoplasm, cytosol in the cellular components, identical protein binding, enzyme binding, and protein kinase activity in the molecular functions. KEGG pathway analysis highlighted pathways like HIF-1 signaling pathway, PI3K-Akt signaling pathway, and AA metabolism as key pathways involved in the therapeutic effects GJC aqueous extract on PF.

### 2.9. Integrating Serum Metabolomics and Network Analyses

To obtain an integrated vision of the mechanisms of action of GJC aqueous extract on PF, network analysis and metabolomics were integrated to set up an interaction network. MetScape was utilized to incorporate metabolites with the potential targets shown in the network pharmacology into a “metabolite–reaction–enzyme–gene” network (Figure 8A), which allowed for the identification of five pivotal targets: CYP2C9, PTGS1, PTGS2, PLA2G2A, and ALOX5. Furthermore, the 122 core targets identified from the network pharmacology and 32 differential metabolites recognized from the metabolomics analysis were imported into the Joint Pathway Analysis section of MetaboAnalyst. As shown in Figure 8B, metabolic pathways including AA metabolism and linoleic acid metabolism with distinct metabolites and genes were also significantly enriched from the network analysis. Notably, AA and linoleic acid metabolism play a crucial role in inflammatory responses, during which they are metabolized by enzymes like cyclooxygenase (COX), lipoxygenase (LOX), and cytochrome P450 (CYP450) to produce bioactive eicosanoids. Finally, by integrating network analysis with metabolomics, we input the enriched targets and differential metabolites in the AA metabolic pathway into “Cytoscape 3.9.1” software. We performed a reverse search for key targets to identify components interacting with them, constructing a “compound–target–metabolite–pathway” network. The results indicate that components such as caffeic acid, loliolid, tiliroside, and tellimagrandin II may exert therapeutic effects through the AA pathway, as shown in Appendix A (Figure 8C).

### 2.10. Validation of Arachidonic Acid Metabolic Pathway

Metabolomics and network analyses have shown the key role of the AA metabolic pathway in the anti-PF effect of GJC aqueous extract and suggest the regulation of inflammation to be involved. To further validate this deduction, Western blot analysis was utilized to evaluate its effect on the protein expressions of key enzymes ALOX5, PTGS2, PTGS1, CYP2C9, and PLA2G2A within the AA metabolic pathway identified in the integrated analysis (Appendix A). The results indicate that, compared to the Control group, the protein expressions of these enzymes were significantly upregulated in the BLM group (*p* < 0.05 or *p* < 0.001). However, upon administration of GJC aqueous extract, their upregulated expressions were reversed to different extents (Figure 9A–F). These findings verify the involvement of the AA metabolic pathway in the anti-PF effect of GJC aqueous extract.

## 3. Discussion

PF is considered an irreversible disease [26]. Although most PF patients exhibit a slow and progressive course within a few years due to delayed diagnosis and acute exacerbations caused by complications, some retrospective longitudinal studies have shown that the average life expectancy of these patients after diagnosis is 3–5 years [27]. Currently, FDA-approved PF drugs can delay the decline in lung function, but they neither reverse fibrosis nor significantly improve overall survival [28]. Therefore, there is an urgent need for new strategies and drugs to treat lung injury and fibrosis. Clinically, GJC is used alone or in combination with other drugs to treat respiratory infections and other lung system diseases. Our research has shown that GJC aqueous extract could effectively improve PF. Studies have demonstrated that TGF-β is a major contributor to EMT, activating the cytoplasmic Smad-2/3 complex and stimulating the nuclear translocation of the Smad-2/3/4 complex. The latter binds to binding elements in the promoters of α-SMA, collagen, and other related genes, leading to the synthesis and remodeling of ECM [29]. Lung expression levels of HYP, α-SMA, and TGF-β1 demonstrate that GJC aqueous extract could effectively inhibit BLM-induced ECM deposition to block the formation of myofibroblasts.

Increasing numbers of studies have shown that intratracheal injection of BLM can be used to simulate the pathological mechanisms of PF and evaluate the pharmacological activity of drugs [28,30]. Inflammation and oxidative stress are widely recognized as the core pathological mechanisms of PF [28]. Anti-inflammation and antioxidation are therefore key strategies to prevent or treat lung fibrosis. Markedly decreased levels of TNF-α and IL-6 indicate that GJC aqueous extract could significantly reduce BLM-induced lung inflammation to alleviate BLM-induced damage. Meanwhile, under oxidation stress, the release of TGF-β would be accelerated [31], and inflammatory cells in the lung would be activated, leading to more severe oxidation stress and creating a vicious cycle of fibrosis [31]. In the present study, the decrease in the level of MDA in the lung tissue homogenate and the increase in the level of SOD after the administration of GJC aqueous extract indicate that GJC aqueous extract can reduce the generation of reactive oxygen radicals in vivo, increase the level of antioxidant enzymes, and regulate oxidative stress.

Metabolomics technology involves detecting changes in endogenous components within an organism following exposure to external stimuli, thereby reflecting the intrinsic mechanisms of diseases or drugs [32].The serum metabolome analysis revealed that 32 differential metabolites were associated with GJC aqueous extract treatment of mice with PF. GJC aqueous extract affects major metabolic pathways, including AA metabolic, sphingolipid metabolic, and vitamin B6 metabolic pathways. These metabolic pathways focusing on AA metabolism and sphingolipid metabolism are involved in the regulation of inflammation. Metabolomics studies have indicated that AA metabolism is the metabolic pathway most relevant to the mechanism by which GJC aqueous extract treats PF, characterized by reduced levels of AA, 5-HETE, and 8-HETE in mice. AA metabolism has been proven to be associated with inflammatory processes and can be further metabolized into eicosanoids, which are potent autocrine and paracrine bioactive mediators and are widely involved in various physiological and pathological processes [32,33].

Network analysis has predicted 26 key components and identified TP53, STAT3, and SRC as the core targets affecting pathological processes such as apoptosis and inflammatory responses in PF [34]. This reveals that the HIF-1 signaling pathway, PI3K-AKT signaling pathway, TNF signaling pathway, and AA metabolic pathway may be key pathways involved in the treatment of PF mediated by GJC aqueous extract. The activation of the PI3K-Akt signaling pathway and TNF signaling pathway can release inflammatory factors such as TNF-α and IL-6 [35,36,37], which are involved in the formation of PF, indicating that GJC aqueous extract may regulate numerous inflammation-related signaling pathways.

The pathology of PF is complex and still unclear. In this study, metabolomics and network analyses were conducted to elucidate the pathways crucial for developing effective treatments for fibrotic lung diseases. The results show that GJC aqueous extract exerts its anti-PF effects mainly by regulating expressions of metabolites (e.g., arachidonic acid, 5-HETE, 8-HETE) and key enzymes (e.g., ALOX5, PTGS2, PTGS1, CYP2C9, PLA2G2A) in the AA metabolic pathway. Furthermore, by integrating network analysis with metabolomics, this study further explored the potential material basis of the intervention of GJC aqueous extract in PF and identified key components of GJC aqueous extract involved in this intervention. Compounds in GJC, such as caffeic acid, loliolid, tiliroside, and tellimagrandin II, may exert therapeutic effects by targeting key enzymes (e.g., ALOX5, PTGS2, PTGS1, CYP2C9, PLA2G2A) in the AA metabolic pathway. Caffeic acid, a hydroxycinnamic acid widely present in fruits and vegetables, has been confirmed to possess anti-inflammatory activity and may potentially enhance the efficacy of aged treatment drugs in controlling pulmonary diseases [38]. Tiliroside, a natural flavonoid glycoside with anti-inflammatory and antioxidant bioactivities, has been shown to inhibit the inflammatory response in lipopolysaccharide (LPS)-stimulated macrophages and the production of cytokines such as TNF-α and IL-6 in LPS-stimulated BV2 microglia [39]. Additionally, tiliroside may combat oxidative stress by selectively modulating the function of antioxidant enzymes, reducing the expression levels of p22phox, COX-2, and NOX4 in LPS-stimulated mouse kidneys [39]. Therefore, key components in GJC may intervene at multiple stages of PF development to improve the symptoms of PF.

Increasing evidence indicates that metabolic dysregulation of AA is closely associated with the pathogenesis of PF [26], and arachidic acid-like metabolites in the pathway of AA have been proven to be critical in the anti-inflammatory, anti-fibrotic, and anti-apoptotic effects of a TCM formula [40]. AA is an important fatty acid that can be metabolized into various bioactive lipid mediators (e.g., prostaglandins, leukotrienes, and HETEs) under normal physiological conditions. Lipids are involved in the pathogenesis and progression of PF by inducing endoplasmic reticulum stress, promoting cell apoptosis, and enhancing the expression of profibrotic biomarkers [41]. 5-HETE, for example, facilitates neutrophil aggregation and migration across barriers, thereby mediating lung inflammation. PLA2G2A, an enzyme in AA metabolism, plays a key role in the fibrotic process by catalyzing the release of arachidonic acid from phospholipid molecules and initiating downstream lipid-mediated inflammatory responses [42,43]. PLA2G2A is typically secreted by damaged lung tissue’s epithelial cells and immune cells [43] and tends to be upregulated during PF. ALOX5 is the enzyme that converts arachidonic acid into leukotrienes capable of stimulating neutrophil recruitment and inflammatory reactions [44]. However, excessive activation of ALOX5 has been shown to worsen inflammatory responses, leading to further lung damage and the advancement of PF [45,46]. PTGS1 and PTGS2 are enzymes involved in prostaglandin synthesis and are activated during acute inflammatory reactions and fibrosis development [46,47]. The actions of PTGS1 and PTGS2 are interconnected, collectively impacting the advancement of lung fibrosis. CYP2C9 was reported to catalyze the production of 11,12-EET, increase the expression of COX-2 [48] in endothelial cells, and show anti-fibrotic effects by promoting the survival of epithelial cells and the death of fibroblasts/myofibroblasts in IPF, suggesting that CYP2C9 may be involved in the pathogenesis of PF [49]. Our study indicates that GJC aqueous extract could inhibit the expression of PLA2G2A, ALOX5, PTGS1, PTGS2, and CYP2C9 to alleviate the inflammatory responses and fibrosis in PF via the AA pathway.

This experiment investigated the mechanism of action of GJC aqueous extract in treating PF using metabolomics and network analysis methods. Initially, in vivo pharmacodynamic validation was conducted, and indices related to inflammation and oxidative stress were measured to assess the overall therapeutic effect of GJC aqueous extract on PF. Subsequently, differential metabolites were obtained through serum metabolomics, and pathway enrichment analysis was performed, showing enrichment of AA metabolism and sphingolipid metabolism. Thereafter, we conducted network analysis on the 26 identified components, determined the common targets of each component and PF, and performed KEGG analysis, identifying key pathways involved in the treatment of PF mediated by GJC aqueous extract as being inflammation-related. By integrating metabolomics and network analyses, we identified metabolic pathways and key targets involved in the pathophysiological processes of PF and performed a reverse search for key targets to determine the components interacting with them, constructing a “compound–target–metabolite–pathway” network. Finally, the above five key targets were validated using Western blotting. In summary, we have summarized the potential targets and effects of some metabolites in GJC extract, as shown in Table 2.

## 4. Materials and Methods

### 4.1. Reagents

BLM (lot number: B802467) was procured from Shanghai Macklin Biochemical Technology (Shanghai, China). PFD (lot number: S80784) was obtained from Shanghai Yuanye Bio-Technology (Shanghai, China). HYP (lot number: A030-2), MDA (lot number: A003-1-2), and SOD (lot number: A001-3-2) kits were purchased from Nanjing Jiancheng Bioengineering Institute (Nanjing, China). IL-6 (lot number: RX203049M) and TNF-α (lot number: RX202412M) kits were acquired from Ruixin Biotechnology (Quanzhou, China). Antibodies including GAPDH (lot number: 60004-1-lg), PTGS2 (lot number: 12375-1-AP), PTGS1 (lot number: 13393-1-AP), and CYP2C9 (lot number: 16546-1-AP) were all sourced from Proteintech Group in Wuhan, China, while ALOX5 (lot number: DF6681) and PLA2GA (lot number: DF6366) were purchased from Affinity Biosciences (Cincinnati, OH, USA).

### 4.2. Plant Material

*Geum japonicum* var. *chinense* (batch number: 230811), purchased from Jinsha Traditional Chinese Medicine Market in Bijie, Guizhou, has been identified by Professor Liu Shaohuan from Guizhou Medical University as the whole plant of *Geum japonicum* Thunb. var. *chinense*.

### 4.3. Preparation of GJC Aqueous Extract

Dried whole herbs of *Geum japonicum* var. *chinense* (300 g) were added into 3 L water and extracted twice at 80 °C (30 min for each); then, the solution was combined, filtered, and freeze-dried to obtain 60.1 g of extract.

### 4.4. UHPLC-Q-Orbitrap HRMS Analysis

To preliminarily detect potential bioactive chemical substances in the aqueous extract of GJC, a Waters Acquity UPLC HSS T3 column (2.1 × 100 mm, 1.8 μm; Waters, Milford, MA, USA) was employed. The mobile phases consisted of acetonitrile (A) and 0.1% aqueous solution of formic acid (B). The injection volume, flow rate, and column temperature were adjusted to 10 µL, 0.3 mL/min, and 40 °C, respectively. The concentration of A in the solution changed with time as follows: 0–2 min (5%~5% A); 2–42 min (5~95% A); 42–47 min (95~95% A); 47–47.1 min (95~5% A); 47.1–50 min (5~5% A). Following that, the mass spectrometry system used was a Thermo Scientific Q Exactive Focus (Thermo Fisher Scientific, Waltham, MA, USA), utilizing full scan/dd-ms^2^ with simultaneous scanning of positive and negative ions. The capillary voltage was set at 3.0 kV for positive ion mode and 2.5 kV for negative ion mode. The heated electrospray ionization (HESI) source temperature was 320 °C, and the capillary temperature was 350 °C. The sheath gas flow rate was 35 arb, and the auxiliary gas flow rate was 10 arb. The scan range was *m*/*z* 100~1500, with an MS^1^ resolution of 70,000 and an MS/MS resolution of 17,500. The normalized collision energy (NCE) was set at 20, 40, and 60 eV. The investigation utilized Xcalibur 3.0 software (Thermo Fisher Scientific, Waltham, MA, USA) for data processing. Based on the high-resolution mass spectrometry information, the elemental composition and molecular formula of the chromatographic peaks were accurately predicted, with a mass error tolerance of 5 ppm. Additionally, Compound Discoverer 2.0 software, equipped with the mzCloud© and ChemSpider© databases (Thermo Fisher Scientific, Waltham, MA, USA), was employed for auxiliary analysis.

### 4.5. Animals Experiments

Ninety SPF-grade male C57BL/6J mice (6–8 weeks old, weighing 18–20 g) were provided by Changsha Tianqin Biotechnology Co., Ltd. (Changsha, China) [License No. SCXK (Xiang 2022-0011)]. The animal experimental protocol was approved by the Ethics Committee of Guizhou Medical University. The experiment was reviewed and approved by the Animal Care Welfare Committee of GuiZhou Medical University (No.2303451, approved 17 October 2023). The animals were housed in a controlled environment with a temperature of 25 ± 5 °C, relative humidity of 60 ± 5%, and a 12 h light/dark cycle, with free access to food and water. According to the 2020 edition of the *Chinese Pharmacopoeia*, the human dosage of *Geum japonicum* var. *chinense* is prescribed as 9–30 g. This study adopted a daily dosage of 9 g as the basis for calculation. Subsequently, the dosage for mice was determined based on the body surface area conversion formula between humans and mice, and the extraction rate of the GJC aqueous extract [53]. After 7 days of acclimatization, the mice were randomly divided into six groups based on their body weight using a digital randomization method: normal group (Control, n = 15), BLM group (BLM, n = 15), and pirfenidone group (PFD, 273 mg·kg^−1^, n = 15), as well as low-dose (GJCL, 115 mg·kg^−1^, n = 15), medium-dose (GJCM, 230 mg·kg^−1^, n = 15) and high-dose (GJCH, 460 mg·kg^−1^, n = 15) GJC aqueous extract groups. Except for those in the normal group, mice in all groups were administered bleomycin (BLM, 3.5 mg/kg) via intratracheal injection to induce the PF model [54]. On the first day after BLM treatment, drugs were given orally by gavage to mice of the corresponding groups for 21 consecutive days, while mice in the Control and BLM groups received an equal volume of distilled water. One hour after the last administration, mouse serum was collected and centrifuged at 3000 rpm for 10 min at 4 °C to separate the serum, which was then stored at −80 °C. Right lung tissue slices obtained from each mouse were fixed in 4% paraformaldehyde and embedded in paraffin. The remaining lung tissue was stored at −80 °C for subsequent analyses.

### 4.6. Survival Rate and Lung Coefficient

The mouse lung tissues were dissected, washed with saline, and blotted dry with filter paper to remove surface moisture before weighing. The lung coefficient was then calculated using the following formula:Lung coefficient = [(Weight of lung tissue (g)/Body weight (g)) × 100%]

### 4.7. H&E Staining and IHC Staining

The lung tissue was fixed overnight with 4% paraformaldehyde and then cut into 5 μm thin sections for Mason’s Trichrome or H&E (Sigma Aldrich, St. Louis, MO, USA), Masson, and IHC staining according to the instructions. The degree of lung injury was analyzed semi-quantitatively according to the degree of leukocyte infiltration, hemorrhage, edema, and alveolar wall thickening in the sections. The grade of inflammation was on a scale of 0–3, corresponding to none, mild, moderate, or severe inflammation, respectively. The degree of pulmonary fibrosis was evaluated according to the Aschoff score. For IHC analysis, the sections were incubated with antibodies against TGF-β1 (1:500, bs-0086R, Bioss, Woburn, MA, USA) and α-SMA (1:500, bs-10196R, Bioss, Woburn, MA, USA) at 4 °C overnight. Then, the sections were incubated with streptavidin-peroxidase HRP, stained with hematoxylin for 5 min, and observed with an optical microscope. The brown stain was the antibodies’ binding area. ImageJ was used for quantitative evaluation of positive markers (brown area).

### 4.8. Biochemical Indicator Testing

Following the manufacturer’s instructions, ELISA kits were used to evaluate the levels of IL-6 and TNF-α (inflammatory cytokines) in serum. Concurrently, detection kits provided by Nanjing Jiancheng Bioengineering Institute were employed to measure the contents of HYP (Hydroxyproline), SOD (superoxide dismutase), and MDA (malondialdehyde) oxidative stress markers in lung tissues (Nanjing, China).

### 4.9. Plasma Sample Preparation

A total of 150 μL of serum samples was slowly thawed at 4 °C, and an aliquot of each sample was transferred to a pre-cooled methanol/acetonitrile/water solution (2:2:1, *v*/*v*). The mixture was vortexed and then sonicated at 4 °C for 30 min. The obtained solution was then allowed to stand at −20 °C for 10 min. Subsequently, centrifugation was performed at 14,000× *g* for 20 min at 4 °C. The supernatant was vacuum-dried, and the obtained sample was redissolved in 100 μL of acetonitrile/water solution (1:1, *v*/*v*), vortexed, and centrifugated at 14,000× *g* for 15 min at 4 °C to give a supernatant for mass spectrometric analysis.

### 4.10. Metabolomics Analysis

For metabolomics analysis, serum samples were separated using an UHPLC, Thermo Fisher Scientific (Waltham, MA, USA), system equipped with an HILIC chromatography column. The column temperature was maintained at 25 °C with a flow rate of 0.3 mL/min and an injection volume of 2 μL. The mobile phase was 0.1% formic acid in water (A) and 0.1% formic acid in acetonitrile solution (B). The gradient elution conditions were as follows: 0–1.5 min (98–98% B); 1.5–12 min (98–2% B); 12–14 min (2–2% B); 14–14.1 min (2–98% B); 14.1–17 min (98–98% B). Throughout the analysis, samples were kept in a 4 °C autosampler. To minimize signal variability, samples were analyzed in a randomized order. Quality Control (QC) samples were interspersed within the sample queue to assess system stability and data reliability. After separation by UHPLC, serum samples were analyzed by mass spectrometry using a Q Exactive series mass spectrometer (Thermo) in both positive and negative electrospray ionization (ESI) mode. The ESI source and mass spectrometry settings were as follows: nebulizer gas (Gas1) and heated auxiliary gas (Gas2) were both set at 60 psi, sheath gas (CUR) at 30 psi, ion source temperature at 600 °C, and spray voltage (ISVF) at ±5500 V for both modes. The primary mass-to-charge ratio (*m/z*) detection range was 80–1200 Da with a resolution of 60,000 and a scan accumulation time of 100 ms. A segmented acquisition method was used for secondary analysis with a scan range of 70–1200 Da, secondary resolution of 30,000, scan accumulation time of 50 ms, and dynamic exclusion time of 4 s. Raw data were converted into .mzXML format using ProteoWizard 3.0, followed by peak alignment, retention time correction, and peak area extraction using XCMS 4.7 software. Orthogonal partial least squares discriminant analysis (OPLS-DA) was performed to determine the metabolic differences between the samples using SIMCA 14.1 software (Umetrics, Umeaa, Sweden). Variable importance projection (VIP) > 1 and *p* < 0.05 were used as the criteria to screen for differential metabolites, which were then identified by online databases such as HMDB, MassBank, mzCloud, and KEGG.

### 4.11. Network Analysis

For the compounds in GJC aqueous extract identified in Section 4.2, structural information for each compound was obtained using the ChemSpider and PubChem databases. Chemical structures were drawn using InDraw 5.1.0 software and saved in sdf and mol file formats. The relevant targets of these compounds were predicted using the DrugBank (https://www.drugbank.com/, accessed on 11 October 2024), SwissTargetPrediction (www.swisstargetprediction.ch/, accessed on 11 October 2024), and PharmMapper (http://lilab-ecust.cn/pharmmapper/index.html, accessed on 14 October 2024) platforms. Predicted target information was validated against the UniProtKB database (https://www.uniprot.org, accessed on 15 October 2024), specifying targets from “Homo sapiens” and converting all target names to official gene symbols. To identify PF-related targets, searches were conducted on OMIM (www.omim.org, accessed on 17 October 2024) and GeneCards (www.genecards.org, accessed on 17 October 2024) databases, filtering results based on score values to exclude targets below the median score. Duplicate genes were removed, and Venn 2.1.0 (https://bioinfogp.cnb.csic.es/tools/venny/index.html, accessed on 20 October 2024) was used to generate a Venn diagram showing overlapping targets between compounds and PF-related targets. Intersection targets with a network confidence score > 0.9 shown by the STRING database (https://string-db.org/, accessed on 20 October 2024) were imported into Cytoscape 3.9.1 for network topology analysis and core target identification. GO and KEGG enrichment analyses of core targets (*p* < 0.01) were performed using the DAVID database (https://david.ncifcrf.gov/, accessed on 25 October 2024). Pathways were ranked based on *p*-values, and the top 20 pathways were visualized using Cytoscape 3.9.1 to construct a “compound–target–pathway” network diagram illustrating interactions between compounds, targets, and pathways related to PF and GJC aqueous extract.

### 4.12. Integrating Serum Metabolomics and Network Analyses

To further investigate the metabolic regulatory mechanisms of GJC aqueous extract, the differential metabolites obtained from Section 4.10 and the core targets identified in Section 4.11 were simultaneously imported into the Joint Pathway Analysis module of the MetaboAnalyst database to establish the metabolite–target–pathway relationships, thereby identifying the crucial metabolic pathways and core targets involved.

### 4.13. Western Blotting

Based on the serum metabolomics and network analyses, Western blotting validation was performed to investigate the proteins involved in the core metabolic pathways. Lung tissue proteins were extracted, and their concentrations were determined using the BCA method. Subsequently, proteins were quantified, denatured, separated by electrophoresis, transferred onto a membrane, and blocked. The membrane was then incubated overnight at 4 °C with primary antibodies of PTGS1 (1:1000), PTGS2 (1:1000), ALOX-5 (1:1000), CYP2C9 (1:2000), PLA2G2A (1:2000), and GAPDH (1:5000). After washing with PBS (pH 7.4), the membrane was incubated with either rabbit or mouse secondary antibodies (1:5000) for 2 h. After an additional wash, chemiluminescent reagents were applied to visualize protein bands, which were then imaged and scanned. Semi-quantitative analysis of band intensities was conducted using ImageJ 1.51 software.

### 4.14. Statistical Analysis

All data were analyzed using Analysis of Variance (ANOVA). The results are presented as the mean ± standard error of the mean (mean ± SEM). Statistical significance was set at a *p*-value less than 0.05 (*p* < 0.05), and analyses were performed using SPSS 22.0 software (IBM, Armonk, NY, USA).

## 5. Conclusions

In summary, GJC aqueous extract exerts potent therapeutic effects on lung inflammation and fibrosis induced by BLM. The regulation of AA metabolism by inhibiting the expression of its key enzymes and metabolites was involved in the anti-PF effect of GJC aqueous extract (Figure 10, by Figdraw 2.0). GJC may be a promising botanical drug for the development of novel anti-PF drugs.

## Figures and Tables

**Figure 1 ijms-26-01462-f001:**
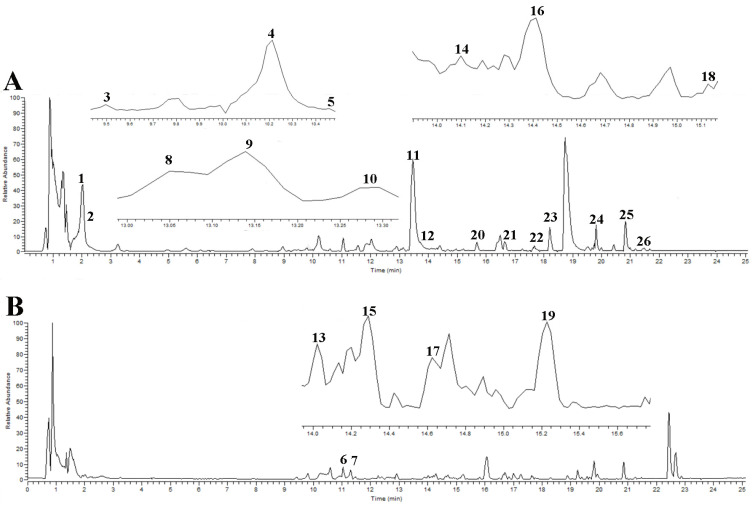
The BPC of GJC aqueous extract under negative (**A**) and positive (**B**) ion mode.

**Figure 2 ijms-26-01462-f002:**
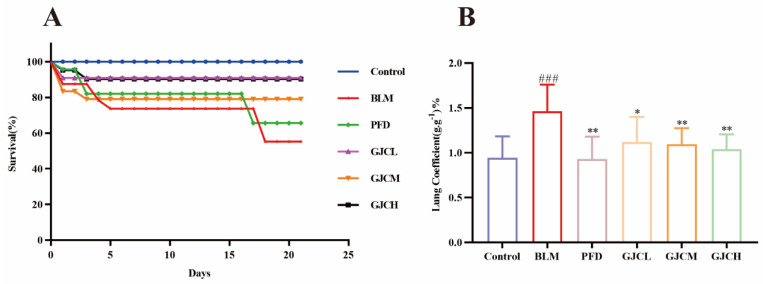
The effects of GJC aqueous extract on the survival rate and lung coefficient of BLM-induced PF mice. (**A**) The survival rate of mice; (**B**) the lung coefficient of mice. All data are shown as the mean ± SEM (n ≥ 8). ^###^
*p* < 0.001 vs. Control; * *p* < 0.05 and ** *p* < 0.01 vs. BLM.

**Figure 3 ijms-26-01462-f003:**
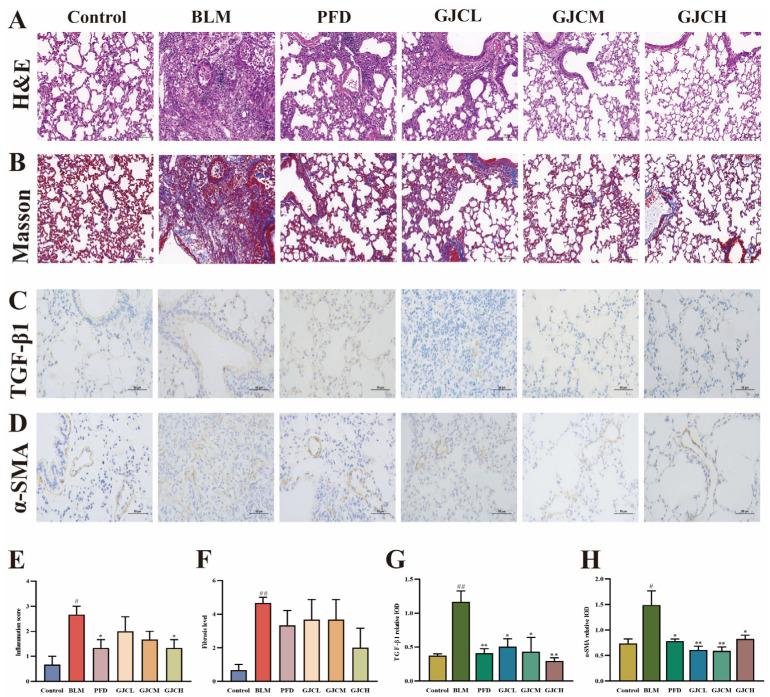
Histopathological and immunohistochemical examinations of mouse lung tissues. Representative images of H&E staining ((**A**), ×400) and corresponding score (**E**); representative images of Masson dyeing ((**B**), ×400) and corresponding score (**F**). Analysis of expression levels of TGF-β1 (**C**,**G**) and α-SMA (**D**,**H**) proteins in the lung sections of mice in all groups. The values represent the mean ± SEM (n = 3). ^#^
*p* < 0.05 and ^##^
*p* < 0.01 vs. Control; * *p* < 0.05 and ** *p* < 0.01 vs. BLM.

**Figure 4 ijms-26-01462-f004:**
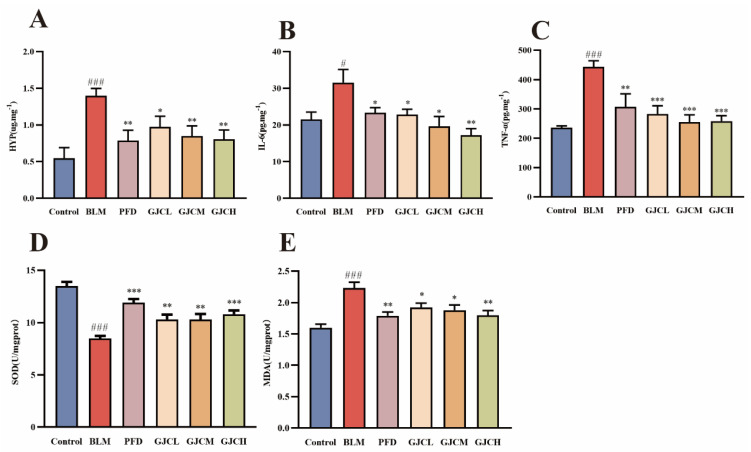
The effect of GJC aqueous extract on inflammatory levels in the serum and HYP level and oxidative stress levels in lung tissue of BLM-induced PF mice. (**A**) HYP level of mice; (**B**,**C**) the expression levels of IL-6 and TNF-α in the serum of mice in all groups; (**D**,**E**) the content of SOD and MDA in the lung tissue of mice in all groups. The values represent the mean ± SEM (n = 8). ^#^
*p* < 0.05 and ^###^
*p* < 0.001 vs. Control. * *p* < 0.05, ** *p* < 0.01, and *** *p* < 0.001 vs. BLM group.

**Figure 5 ijms-26-01462-f005:**
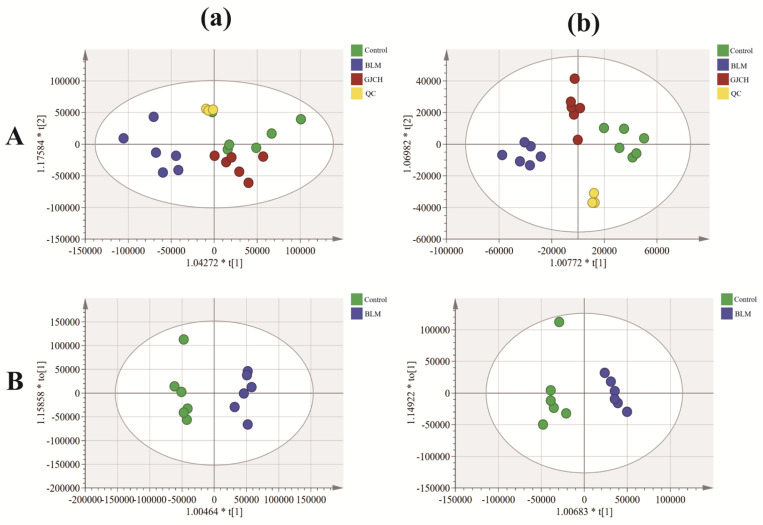
Multivariate statistical analysis results for the serum samples (n = 6). (**a**) Negative ion mode and (**b**) positive ion mode. (**A**) OPLS-DA analysis of the QC, Control, BLM, and GJCH groups; (**B**) OPLS-DA analysis of the Control and BLM groups.

**Figure 6 ijms-26-01462-f006:**
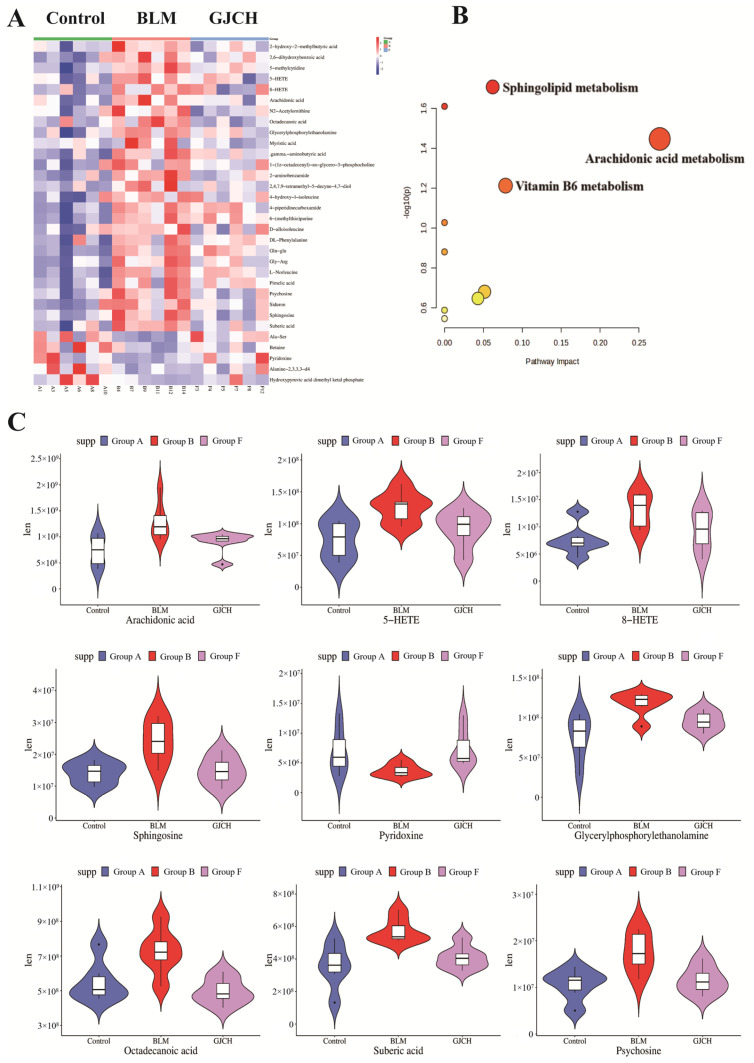
Heat map and metabolic pathway analysis of serum metabolites. (**A**) Heatmap of 32 differentially expressed metabolites in serum; (**B**) enrichment analysis of differentially expressed metabolites in serum (The size of the circle is usually related to the Pathway Impact Value of the metabolite. The larger the circle, the higher the impact value of the metabolite in the pathway. The color indicates the significance level (*p*-value) of the metabolite, with darker colors representing smaller *p*-values.); (**C**) expression levels of 9 selected metabolites with significant differences. The small black dots may represent outliers in a dataset.

**Figure 7 ijms-26-01462-f007:**
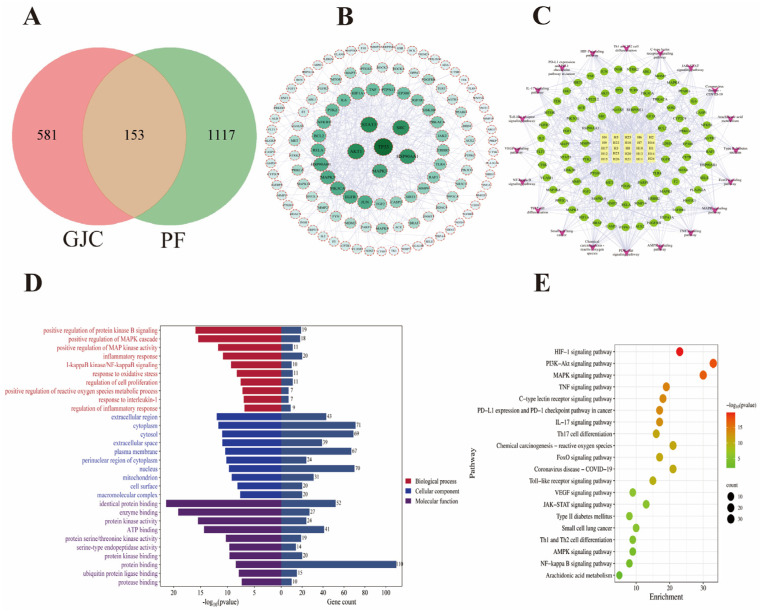
Network analysis of GJC aqueous extract in improving PF. (**A**) Venn diagram of shared targets between GJC aqueous extract and PF; (**B**) protein–protein interaction network; (**C**) compound–target–pathway network of GJC in improving PF; (**D**) GO analysis; (**E**) KEGG pathway analysis.

**Figure 8 ijms-26-01462-f008:**
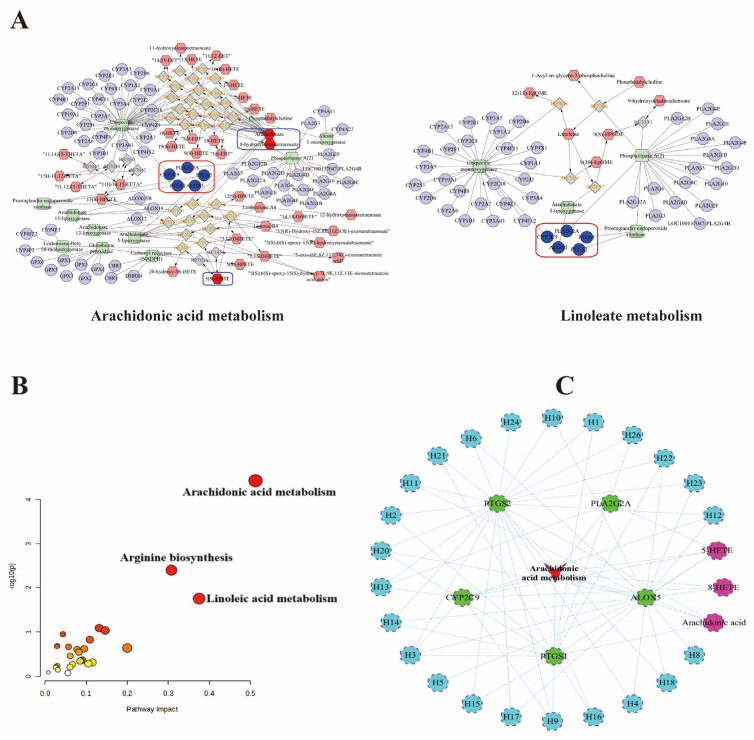
Comprehensive analysis of metabolomics and network pharmacology. (**A**) “Potential metabolite–reaction–enzyme–gene” interaction network. Red hexagons represent metabolites, gray diamonds represent reactions, green rectangles represent enzymes, purple circles represent genes, and blue circles represent core genes. (**B**) Enrichment analysis of metabolite–target network. The larger the circle, the higher the impact value of the metabolite in the pathway. The color indicates the significance level (*p*-value) of the metabolite, with darker colors representing smaller *p*-values. (**C**) “Compound–target–metabolites–AA metabolic pathway” network. The size of the circle is usually related to the Pathway Impact Value of the metabolite.

**Figure 9 ijms-26-01462-f009:**
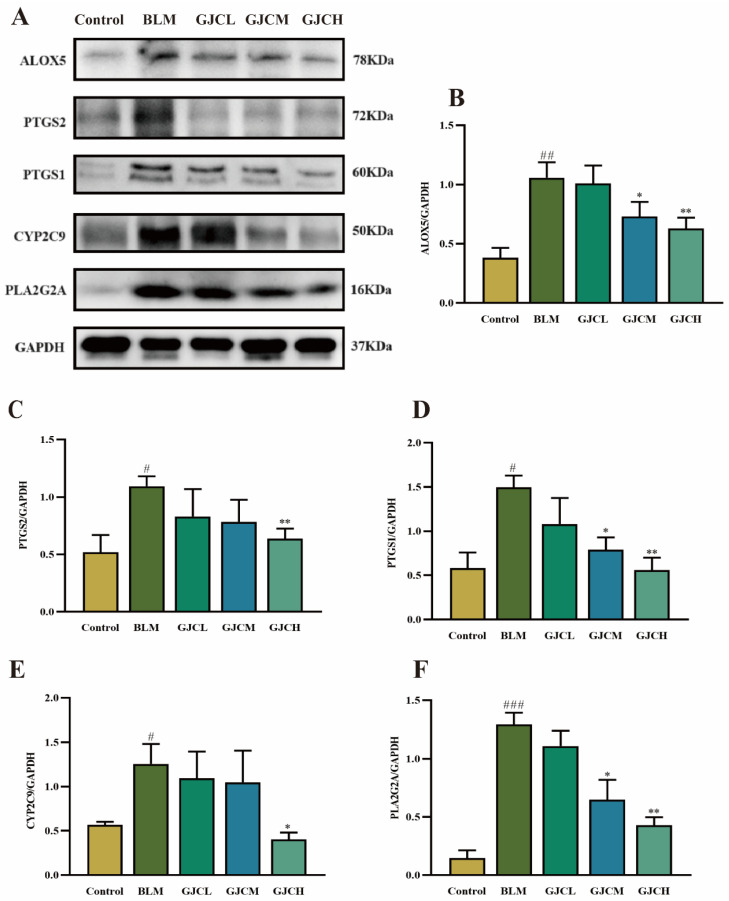
The effect of GJC aqueous extract on the AA metabolic pathway in PF mice. (**A**) The protein bands of ALOX5, PTGS2, PTGS1, CYP2C9, PLA2G2A, and GAPDH, (**B**–**F**) The expression of ALOX5, PTGS2, PTGS1, CYP2C9, and PLA2G2A in lung tissues. The values represent the mean ± SEM (n = 3). ^#^
*p* < 0.05, ^##^
*p* < 0.01 and ^###^
*p* < 0.001 vs. Control group. * *p* < 0.05 and ** *p* < 0.01 vs. BLM.

**Figure 10 ijms-26-01462-f010:**
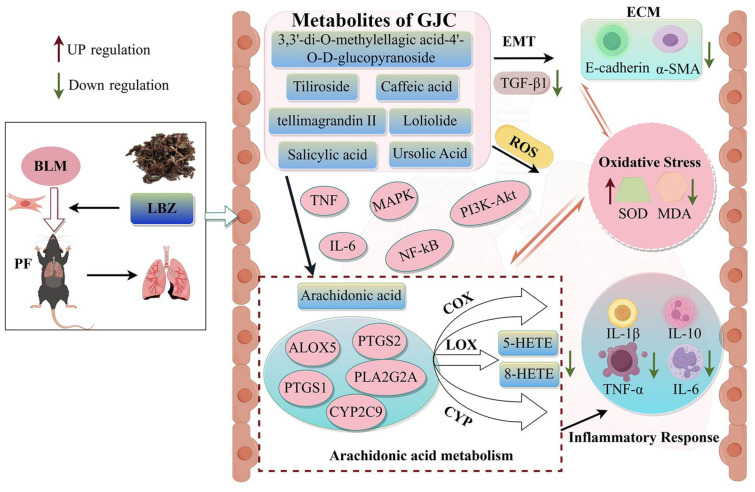
Schematic diagram of mechanism of GJC aqueous extract in BLM-induced PF mice.

**Table 1 ijms-26-01462-t001:** Identification of compounds present in GJC aqueous extract.

NO	Compound	Molecular Formula	t_R_ (min)	Precursor Ion	Ion Mode	Error (ppm)	Product Ion	Reference
1	5-Hydroxymethyl furfural	C_6_H_6_O_3_	2.01	125.0231	[M−H]^−^	−1.76	107.0125	-
2	Gallic acid	C_7_H_6_O_5_	2.23	169.0301	[M−H]^−^	5.5	125.0231	[15]
3	Caffeic acid	C_9_H_8_O_4_	9.50	179.0339	[M−H]^−^	0.19	135.0439, 91.0549	[16]
4	Sanguiin H-4	C_27_H_22_O_18_	10.21	633.0726	[M−H]^−^	−4.9	300.9986, 169.0132	[17]
5	Casuarinin or Potentillin	C_41_H_28_O_26_	10.46	935.0764	[M−H]^−^	−2.31	633.0725, 300.9985, 275.0194	[18]
6	Kaempferol	C_15_H_10_O_6_	11.05	287.0540	[M+H]^+^	−3.46	231.0650, 213.0540, 153.0178	[19]
7	Vanillin	C_8_H_8_O_3_	11.30	153.0544	[M+H]^+^	−1.64	nd	[15]
8	p-Coumaric acid	C_9_H_8_O_3_	13.05	163.0.89	[M−H]^−^	−0.37	119.0489	[16]
9	Tellimagrandin II	C_41_H_30_O_26_	13.14	937.0945	[M−H]^−^	0.34	785.0825, 465.1026, 300.9984	[20]
10	Quercetin 3-O-β-D-Glucuronide	C_21_H_18_O_13_	13.30	477.0660	[M−H]^−^	−0.68	301.0351, 169.0125	[15]
11	Ellagic acid	C_14_H_6_O_8_	13.45	300.9985	[M−H]^−^	2.15	283.9958, 257.0086	[15]
12	3,3′-di-O-methylellagic acid-4′-O-β-D-glucopyranoside	C_22_H_20_O_13_	13.97	491.0822	[M−H]^−^	−1.81	476.0577, 312.9985, 285.0035, 257.0083	[21]
13	Loliolide	C_11_H_16_O_3_	14.02	197.1169	[M+H]^+^	−1.78	179.1062, 161.0124	[22]
14	Salicylic acid	C_7_H_6_O_3_	14.10	137.0232	[M−H]^−^	−1.24	93.0332	[15]
15	Cupressoside A	C_25_H_32_O_10_	14.29	493.2054	[M+H]^+^	−2.93	409.1625, 297.1007	[23]
16	4-Hydroxybenzoic acid	C_7_H_6_O_3_	14.41	137.0231	[M−H]^−^	−1.61	93.0331	-
17	Astragalin	C_21_H_20_O_11_	14.62	449.1064	[M+H]^+^	−3.25	287.0542, 259.0960, 231.1009	[15]
18	(7*S*,8*S*)-5-methoxycupressoside A	C_26_H_34_O_11_	15.13	521.2020	[M−H]^−^	0.31	491.2125	[23]
19	Germacrone	C_15_H_22_O	15.23	219.1737	[M+H]^+^	−2.06	201.1633, 137.1322, 121.1011	-
20	3-O-Methylellagic acid	C_15_H_8_O_8_	15.65	315.0142	[M−H]^−^	1.97	299.9907, 282.9892	-
21	Niga-ichigoside F1	C_36_H_58_O_11_	16.65	711.3956	[M−H+HCOOH]^−^	0.76	503.3371, 485.3264, 421.3069	[24]
22	Tiliroside or cis-Tiliroside	C_30_H_26_O_13_	17.68	593.1287	[M−H]^−^	−0.42	285.0399, 255.0292, 227.0341	[15]
23	3,3′-di-O-methylellagic acid	C_16_H_10_O_8_	18.21	329.0297	[M−H]^−^	1.51	314.0063, 298.9928, 283.9951	[21]
24	1, 2, 19-Trihydroxy-3-oxo-12-ursen-28-oic acid	C_30_H_46_O_6_	20.19	503.3353	[M+H]^+^	−2.87	485.3252, 439.3195, 421.3093	[24]
25	3′-*O*-methyl-3,4-methylenedioxo ellagic acid	C_16_H_8_O_8_	20.83	327.0139	[M−H]^−^	1.06	311.9906, 283.9955, 256.0005, 240.0056	[17]
26	19α-Hydroxyasiatic acid	C_30_H_48_O_6_	21.48	503.3372	[M−H]^−^	0.94	485.3267, 421.3115	[24]

No detected (nd) indicates that no secondary fragment ions were detected.

**Table 2 ijms-26-01462-t002:** Potential targets and effects of some metabolites in GJC extract.

Compound	Target	The Relationship with Arachidonic Acid Metabolism	Reference
Caffeic acid	CYP2C9	Generation of epoxyeicosatrienoic acids (EETs); anti-inflammatory and antioxidant effects	[48,49]
	PTGS1, PTGS2	Inhibition of prostaglandin production; anti-inflammatory and immune regulation	[48,50]
	ALOX5	Inhibition of leukotriene production; anti-inflammatory	[41,51]
Loliolid	PLA2G2A	Release of arachidonic acid; involved in inflammatory responses	[42,43]
	PTGS2	Generation of prostaglandins, which may exacerbate pulmonary fibrosis	[46,47]
	CYP2C9	The production of 20-HETE may influence vasoconstriction and inflammation	[48,49]
Tiliroside	PTGS1, PTGS2	Inhibition of prostaglandin production; anti-inflammatory	[46,47]
	ALOX5	Inhibition of leukotriene production; anti-inflammatory	[44,45,46]
Tellimagrandin II	PTGS1, PTGS2	Inhibition of prostaglandin production; anti-inflammatory, and mediating ferroptosis	[51]
	CYP2C9	The production of 20-HETE may influence vasoconstriction and inflammation	[48,49]
3,3′-di-O-methylellagic acid-4′-O-β-D-glucopyranoside	CYP2C9	The production of 20-HETE may influence vasoconstriction and inflammation	[48,49]
	PTGS2	Inhibition of prostaglandin production; anti-inflammatory	[46,47]
	ALOX5	Inhibition of leukotriene production; anti-inflammatory	[44,45,46]
3-O-Methylellagic acid	CYP2C9	The production of 20-HETE may influence vasoconstriction and inflammation	[48,49]
	PTGS2	Inhibition of prostaglandin production; anti-inflammatory	[46,47]
	ALOX5	Inhibition of leukotriene production; anti-inflammatory	[44,45,46]
Salicylic acid	PTGS1, PTGS2	Inhibition of prostaglandin production; anti-inflammatory	[52]
	ALOX5	Inhibition of leukotriene production; anti-inflammatory	[52]

## Data Availability

The datasets supporting the conclusions of this article are included within the article.

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
