# Peer review of "Integrating Metabolomics and Network Analyses to Explore Mechanisms of Geum japonicum var. chinense Against Pulmonary Fibrosis: Involvement of Arachidonic Acid Metabolic Pathway"

_ijms, 2025, doi:10.3390/ijms26041462_

Round 1

Reviewer 1 Report

Comments and Suggestions for Authors

The manuscript entitled Integrating serum metabolomics and network analysis to explore the mechanisms of Geum japonicum. var. chinense aqueous extract against Pulmonary fibrosis: Involvement of arachidonic acid metabolism pathway’ is interesting and organized. However, the authors need to make some changes.

-       The title is long. Also keywords.

-       Please check the abbreviations throughout the manuscript. It would be best to introduce the acronym when the whole word appears the first time in the text and then use only the abbreviations.

-       Please highlight and emphasize the novelty of this study.

-       Please simplify the graphical abstract

-       Why did the authors choose an aqueous extract of the plant, not an alcoholic one?

-       Cite a relevant reference for in vivo studies, especially for doses.

-       Remove this sentence: Page 3, Line 93,  Table 1. This is a table. Tables should be placed in the main text near to the first time they are cited.

-       Please, add Table 1 Legend.  

-       The representation of Table 1 needs improvements and adjustments as the cells of compounds shift.

-       Please revise the Molecular Formula in Table 1.

-       Please clarify why Compounds 25 and 26 take these numbers, although they have less retention time than compounds 21 to 24.

- Please use clear, darker colors in Figure 7 (D) GO analysis.

-       Authors need to correct some grammatical mistakes.

Comments on the Quality of English Language

-       Authors need to correct some grammatical mistakes.

Author Response

Comments 1: [The title is long. Also keywords.]

Response 1:  [We consider amending the original title to: ‘Integrating metabolomics and network analysis to explore the mechanisms of Geum japonicum. var. chinense against Pulmonary fibrosis: Involvement of arachidonic acid metabolism pathway’.The keywords are modified as ‘Geum japonicum. var. chinense, Pulmonary fibrosis, Metabolomics, Network analysis, Arachidonic acid metabolism’.] Thank you for pointing this out. We agree with this comment. Therefore, We have made revisions to Page 1, Line 2-4, 36, and 37

“[The text in the manuscript has been updated]”

Comments 2: [Please check the abbreviations throughout the manuscript. It would be best to introduce the acronym when the whole word appears the first time in the text and then use only the abbreviations.]

Response 2: Thank you for your valuable suggestion. In this manuscript, all instances where“Pulmonary fibrosis”should be abbreviated as“PF”have been corrected (Line 24, 76, 160, 269, 305, 326, 373, 374, and 519). Similarly, all instances where“arachidonic acid”should be abbreviated as“AA”have also been corrected(Line 35, 37, 190, 191, 220,236, 241, 237, 241, 252, 254, 258, 263, 265, 301, 303, 304, 306, 313, 324, 329, 371, and 557).]

“[The text in the manuscript has been updated]”

Comments 3: [Please highlight and emphasize the novelty of this study]

Response 3: Thank you for your valuable suggestion.The novelty of this study is as follows:

  • GJC aqueous extract mitigates bleomycin-induced lung tissue injury.
  • GJC aqueous extract treatment PF potential mechanism is by inhibiting inflammation response and oxidative stress levels.

GJC aqueous extract may exert anti-pulmonary fibrosis effects by modulating the arachidonic acid metabolic pathway.

Comments 4: [Please simplify the graphical abstract]

Response 4: Thank you for your valuable suggestion.The revised graphical abstract is as follows:

“[The text in the manuscript has been updated]”

Comments 5: [Why did the authors choose an aqueous extract of the plant, not an alcoholic one]

Response 5: Thank you for your inquiry. [In this study, we have opted to use aqueous extracts to emulate the traditional method of preparing and consuming traditional Chinese medicine (TCM) through decoction. This approach is particularly suitable for TCM that requires prolonged administration and possesses mild medicinal properties. Moreover, aqueous extracts offer several advantages, including high extraction efficiency, preservation of medicinal efficacy, ease of application, and broad applicability. These benefits render aqueous extracts irreplaceable in the research of TCM. They not only reflect the traditional value of water extraction but also align with the demands of modern scientific research, providing a solid foundation for the modernization and application of TCM.]

Comments 6: [Cite a relevant reference for in vivo studies, especially for doses]

Response 6: Thank you for your valuable suggestion. [The relevant references for the doses of animals in the in vivo studies have been attached in the text [50] and [51]. We have cited the references on Page 16, Line 435 and 441.]

Comments 7: [Remove this sentence: Page 3, Line 93, Table 1. This is a table. Tables should be placed in the main text near to the first time they are cited’. And please, add Table 1 Legend.]

Response 7: Thank you for your kind reminder. [I deeply apologize for forgetting to delete this sentence when copying the template. It has now been removed, and Table 1 has been renamed to: Identification of compounds present in GJC aqueous extract.]

Comments 8: [The representation of Table 1 needs improvements and adjustments as the cells of compounds shift.]

Response 8: Thank you for your kind reminder. [Table 1 has been revised in the text (page 4, Line 99).]

“[The text in the manuscript has been updated]”

Comments 9: [Please revise the Molecular Formula in Table 1.]

Response 9: Thank you for your kind reminder. [I’m really sorry for this mistake. In Table 1, the molecular formula of the Sanguiin H-4 has been revised to C27H22O18 (page 4, Line 99).]

“[The text in the manuscript has been updated]”

Comments 10: [Please clarify why Compounds 25 and 26 take these numbers, although they have less retention time than compounds 21 to 24.]

Response 10: Thank you very much for your attention. [I sincerely apologize for the oversight in the details previously. Initially, I only analyzed and identified 24 compounds, which were sequentially numbered from 1 to 24 based on their retention times. Later, building on this foundation, we further identified two additional compounds. These were also named according to their retention times as 25 and 26. It is important to note that we did not re-sort these two newly discovered compounds together with the original 24 compounds. Therefore, although the retention times of compounds 25 and 26 may be shorter than those of compounds 21 and 24, they were still named as 25 and 26 according to the order in which they were discovered.] Therefore, to ensure the accuracy and logic of the data and to avoid any potential misunderstandings, we have decided to adjust Table 1 by reordering the compounds according to their retention times from smallest to largest. At the same time, we have thoroughly checked and corrected the information related to the compounds in the text, including Figure S1 in the supplementary materials and Figures 1A, 7C and 8C in the main text, to ensure the consistency and accuracy of all information.

Legends:

Table 1  Identification of compounds present in GJC aqueous extract.

Figures 1  The BPC of GJC aqueous extract under negative (A) and positive (B) ion mod.

Figures 7  Network analysis of GJC aqueous extract in improving PF. (A) Venn diagram of shared targets between GJC aqueous extract and PF; (B) Protein‒protein interaction network; (C) Com-pounds-Target-Pathway network of GJC on improving PF; (D) GO analysis; (E) KEGG pathway analysis.

Figures 8  Comprehensive analysis of metabolomics and network pharmacology. (A) “Potential metabo-lite-reaction-enzyme-gene’’ interaction network. Red hexago-ns represent metabolites, gray dia-monds represent reactions, green rectangles represent enzymes, purple circles represent genes and blue circles represent core genes; (B) Enrichment analysis of metabolite-target; (C) " Com-pounds-Targets-Metabolites- AAmetabolic pathway" network.

NO

Compounds

Molecular formula

tR (min)

Precursor ion

Ion mode

Error (ppm)

Product ions

Reference

1

5-Hydroxymethyl furfural

C6H6O3

2.01

125.0231

[M -H]-

-1.76

107.0125

-

2

Gallic acid

C7H6O5

2.23

169.0301

[M -H]-

5.5

125.0231

[15]

3

Caffeic acid

C9H8O4

9.50

179.0339

[M -H]-

0.19

135.0439、91.0549

[16]

4

Sanguiin H-4

C27H22O18

10.21

633.0726

[M -H]-

-4.9

300.9986、169.0132

[17]

5

Casuarinin or Potentillin

C41H28O26

10.46

935.0764

[M -H]-

-2.31

633.0725、300.9985、275.0194

[18]

6

Kaempferol

C15H10O6

11.05

287.0540

[M+H]+

-3.46

231.0650、213.0540、153.0178

[19]

7

Vanillin

C8H8O3

11.30

153.0544

[M+H]+

-1.64

nd

[15]

8

p-Coumaric acid

C9H8O3

13.05

163.0.89

[M -H]-

-0.37

119.0489

[16]

9

Tellimagrandin II

C41H30O26

13.14

937.0945

[M -H]-

0.34

785.0825、465.1026、300.9984

[20]

10

Quercetin 3-O-β-D-Glucuronide

C21H18O13

13.30

477.0660

[M -H]-

-0.68

301.0351、169.0125

[15]

11

Ellagic acid

C14H6O8

13.45

300.9985

[M -H]-

2.15

283.9958、257.0086

[15]

12

3,3'-di-O-methylellagic acid-4'-O-b-D-glucopyranoside

C22H20O13

13.97

491.0822

[M -H]-

-1.81

476.0577、312.9985、285.0035、257.0083

[21]

13

Loliolide

C11H16O3

14.02

197.1169

[M+H]+

-1.78

179.1062、161.0124

 [22]

14

Salicylic acid

C7H6O3

14.10

137.0232

[M -H]-

-1.24

93.0332

 [15]

15

Cupressoside A

C25H32O10

14.29

493.2054

[M+H]+

-2.93

409.1625、297.1007

 [23]

16

4-Hydroxybenzoic acid

C7H6O3

14.41

137.0231

[M -H]-

-1.61

93.0331

-

17

Astragalin

C21H20O11

14.62

449.1064

[M+H]+

-3.25

287.0542、259.0960、231.1009

 [15]

18

(7S,8S)-5-methoxycupressoside A

C26H34O11

15.13

521.2020

[M -H]-

0.31

491.2125

 [23]

19

Germacrone

C15H22O

15.23

219.1737

[M+H]+

-2.06

201.1633、137.1322、121.1011

-

20

3-O-Methylellagic acid

C15H8O8

15.65

315.0142

[M -H]-

1.97

299.9907、282.9892

-

21

Niga-ichigoside F1

C36H58O11

16.65

711.3956

[M-H+ HCOOH]-

0.76

503.3371、485.3264、421.3069

[24]

22

Tiliroside or cis-Tiliroside

C30H26O13

17.68

593.1287

[M -H]-

-0.42

285.0399、255.0292、227.0341

[15]

23

3,3'-di-O-methylellagic acid

C16H10O8

18.21

329.0297

[M -H]-

1.51

314.0063、298.9928、283.9951

[21]

24

1, 2, 19-Trihydroxy-3-oxo-12-ursen-28-oic acid

C30H46O6

20.19

503.3353

[M+H]+

-2.87

485.3252、439.3195、421.3093

[24]

25

3'-O-methyl-3,4-methylenedioxo ellagic acid

C16H8O8

20.83

327.0139

[M -H]-

1.06

311.9906、283.9955、256.0005、240.0056

[17]

26

19α-Hydroxyasiatic acid

C30H48O6

21.48

503.3372

[M -H]-

0.94

485.3267、421.3115

[24]

“[The text in the manuscript has been updated]”

Comments 11: [Please use clear, darker colors in Figure 7 (D) GO analysis.]

Response 11: Thank you for your kind reminder.The revised image is as follows:

Thank you very much for your attention and time. Looking forward to hearing from you.

Thank you for your consideration.

With my best regards,

Xun He, Prof.

Guizhou Medical University, Guizhou, China

Reviewer 2 Report

Comments and Suggestions for Authors

In my oppinion, the manuscript is very interesting, and it also attempts to improve knowledge about the effects of new therapies against diseases that seriously affect humans. Some considerations:

- In point 2.2, before talking about "HYP", the first paragraph of 2.4 should have been included, where it is explained what this abbreviation is. Be careful with abbreviations and put what they are first in the text.

- In figure 7. D, the legends are barely readable, see if the image quality can be increased.

- In the discussion point, the possible targets of the metabolites identified in the GJC extract are discussed, perhaps putting a table with these metabolites and their possible effects, both those seen in the metabolism of arachidonic acid and the possible effects detected in the literature, so that a better idea of ​​the scope of this extract not studied until now would be given.

- In the references, it would be appreciated to put the DOI.

Author Response

Comments 1: [In point 2.2, before talking about "HYP", the first paragraph of 2.4 should have been included, where it is explained what this abbreviation is. Be careful with abbreviations and put what they are first in the text.]

Response 1: [Thank you for your kind reminder. I’m really sorry for this mistake. In point 2.2, HYP is redundant and has now been removed.(page 5 , Line 104)]

“[The text in the manuscript has been updated]”

Comments 2: [In figure 7. D, the legends are barely readable, see if the image quality can be increased.]

Response 2: Thank you for your kind reminder. We have done our best to improve the image quality of Figure 7. D. The revised image is as follows:

“[The text in the manuscript has been updated]”

Comments 3: [In the discussion point, the possible targets of the metabolites identified in the GJC extract are discussed, perhaps putting a table with these metabolites and their possible effects, both those seen in the metabolism of arachidonic acid and the possible effects detected in the literature, so that a better idea of the scope of this extract not studied until now would be given.]

Response 3: Thank you for your valuable suggestion. According to your suggestion, we have added Table 4 "Potential Targets and Effects of Some Metabolites in GJC Extract" on Page 14, Line 581 of the manuscript.

Table 4 Potential Targets and Effects of Some Metabolites in GJC Extract.

Compounds

Target

The relationship with arachidonic acid metabolism

Reference

Caffeic acid

CYP2C9

Generation of epoxyeicosatrienoic acids (EETs), anti-inflammatory and antioxidant effects

[48, 49]

PTGS1, PTGS2

Inhibition of prostaglandin production, anti-inflammatory and immune regulation.

[48, 52]

ALOX5

Inhibition of leukotriene production, anti-inflammatory

[41, 53]

Loliolid

PLA2G2A

Release of arachidonic acid, involved in inflammatory responses

[42, 43]

PTGS2

Generation of prostaglandins, which may exacerbate pulmonary fibrosis

[46, 47]

CYP2C9

The production of 20-HETE may influence vasoconstriction and inflammation

[48, 49]

Tiliroside

PTGS1, PTGS2

Inhibition of prostaglandin production, anti-inflammatory

[46, 47]

ALOX5

Inhibition of leukotriene production, anti-inflammatory

[44-46]

Tellimagrandin II

PTGS1, PTGS2

Inhibition of prostaglandin production, anti-inflammatory,and mediating ferroptosis

[53]

CYP2C9

The production of 20-HETE may influence vasoconstriction and inflammation

[48, 49]

3,3'-di-O-methylellagic acid-4'-O-β-D-glucopyranoside

CYP2C9

The production of 20-HETE may influence vasoconstriction and inflammation

[48, 49]

PTGS2

Inhibition of prostaglandin production, anti-inflammatory

[46, 47]

ALOX5

Inhibition of leukotriene production, anti-inflammatory

[44-46]

3-O-Methylellagic acid

CYP2C9

The production of 20-HETE may influence vasoconstriction and inflammation

[48, 49]

PTGS2

Inhibition of prostaglandin production, anti-inflammatory

[46, 47]

ALOX5

Inhibition of leukotriene production, anti-inflammatory

[44-46]

Salicylic acid

PTGS1, PTGS2

Inhibition of prostaglandin production, anti-inflammatory

[54]

ALOX5

Inhibition of leukotriene production, anti-inflammatory

[54]

Comments 4: [In the references, it would be appreciated to put the DOI.]

Response 4: Thank you for your valuable suggestion, following your suggestion, we have added the DOI to the references.

Thank you very much for your attention and time. Looking forward to hearing from you.

Thank you for your consideration.

With my best regards,

Xun He, Prof.

Guizhou Medical University, Guizhou, China
